# Structural Properties and Catalytic Activity of Binary Poly (vinyl alcohol)/Al₂O₃ Nanocomposite Film for Synthesis of Thiazoles

**Sayed M. Riyadh** [1,3]**, Khaled D. Khalil** [2,3,*]  **and Ali H. Bashal** [2]

[1]   Department of Chemistry, Faculty of Science, Taibah University, Al-Madinah Almunawrah 30002,
     Saudi Arabia; riyadh1993@hotmail.com
[2]   Department of Chemistry, Faculty of Science, Taibah University, Al-Madinah Almunawrah, Yanbu 46423,
     Saudi Arabia; abishil@taibahu.edu.sa
[3]   Department of Chemistry, Faculty of Science, Cairo University, Giza 12613, Egypt
*   Correspondence: khd.khalil@yahoo.com

**Abstract:** A solution casting technique was applied to prepare a binary poly (vinyl alcohol)/Al₂O₃ nanocomposite. The structural properties of nanocomposite were investigated using Fourier-transform infrared spectra, field emission scanning electron microscope, energy dispersive X-ray analyses, and X-ray diffraction. The hybrid PVA/Al₂O₃ film exhibited a conspicuous catalytic performance for synthesis of thiazole derivatives under mild reaction conditions. Moreover, the optimization of catalytic efficiency and reusability of this nanocomposite have been investigated.

**Keywords:** poly (vinyl alcohol); aluminum oxide; nanocomposite; heterogeneous catalysis; thiazoles

---

## 1. Introduction

Polyvinyl alcohol (PVA) is a biodegradable, water-soluble, and environmentally benign polymeric material with unique emulsifying, adhesive, and film-forming properties [1]. It can be used in the synthesis of organic-inorganic hybrid membrane for biodiesel production [2] and has been regarded as appropriate matrix in microbial fuel cell applications [3]. PVA has also been used in production of biomedical hydrogels with high ability to swell [4]. Moreover, diversity of industrial applications of PVA has been reported such as optical and humidity sensors, food packaging, surgical devices, manufacture of polarizing sheet, and production of thin film transistors [5,6]. Recently, many industrial applications of aluminum oxide (Al₂O₃) nanoparticles have been disseminated such as; drilling fluids agent [7], separators for lithium batteries [8], and biosensors for vitamin E [9]. Aluminum oxide nanoparticles were also used to enhance the thermal and mechanical properties of wood fibers [10]. Conspicuously, aluminum oxide nanoparticles, with large surface area and pore-size distribution, have superior catalytic activities for diverse organic reactions [11,12]. Recently, metal oxide nanoparticles embedded within the polymer matrix have attracted growing interest due to the unique properties that displayed by the hybrid nanocomposites. The properties of the created nanocomposites depend on many factors like the method of preparation, the morphological structure of the prepared nanocomposite, nature of metal oxide nanoparticles that incorporated in, the structural features of the polymer used, and the chemical properties of both metal oxide and polymer molecules. The physicochemical properties of the nanosized particles within the hybrid material differ markedly from those of molecular and bulk materials. In our previous work we have concluded that, the synergistic interaction between polymers and inorganic nanoparticles leads to simulative formation of polymer/inorganic nanocomposite with novel catalytic performance for synthesis of different azoles [13,14]. The preparation of PVA/Al₂O₃ composites has received much attention in the past

few years for various applications [15,16]. The purpose of this study is the preparation of new PVA/Al$_2$O$_3$ thin film nanocomposite by using simple solution casting method (Figure 1). Moreover, the nanocomposite hybrid material was characterized and investigated as powerful recyclable catalyst for synthesis of bioactive azoles.

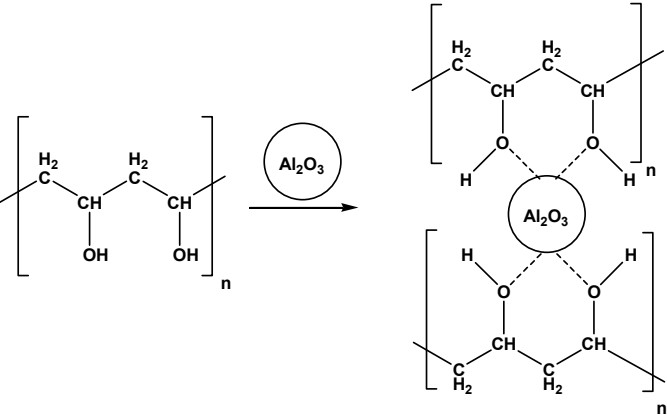

**Figure 1.** Structure of a PVA-Al$_2$O$_3$ nanocomposite.

## 2. Results and Discussion.

### 2.1. Characterization of PVA/Al$_2$O$_3$ Composite Film

#### 2.1.1. Fourier-Transform Infrared (FTIR) Spectra

The FTIR spectra of virgin PVA [17] and PVA-Al$_2$O$_3$ film were measured and the assignment of the most evident absorption bands are shown in Figure 2. The FTIR spectra of virgin PVA revealed an intense and broad stretching band at $v = 3319$ cm$^{-1}$ and another bending vibration at $v = 1634$ cm$^{-1}$ assignable to hydroxyl group. In addition, there is a clear decrease in the intensity of these bands for the hybrid PVA-Al$_2$O$_3$ film, which is attributed to the interaction with Al ions [18] and the influence of the drying temperature. Also, the absorption bands at $v = 2945$ and 1428 cm$^{-1}$ appeared as a result of stretching and bending vibrations of (CH) and (CH$_2$) groups respectively. The broad absorption peak at $v = 1091$ cm$^{-1}$ indicates the C–O stretching vibration [19]. In the finger print region for the spectra of hybrid PVA-Al$_2$O$_3$ film, two characteristic peaks at $v = 627$ and 579 cm$^{-1}$ of Al$_2$O$_3$ [12] are shown as an evidence for the incorporation of Al$_2$O$_3$ molecules in the polymer matrix. Finally, the shift and the intensity decrease in all bands were familiar as result of the interaction between the PVA and metal oxide.

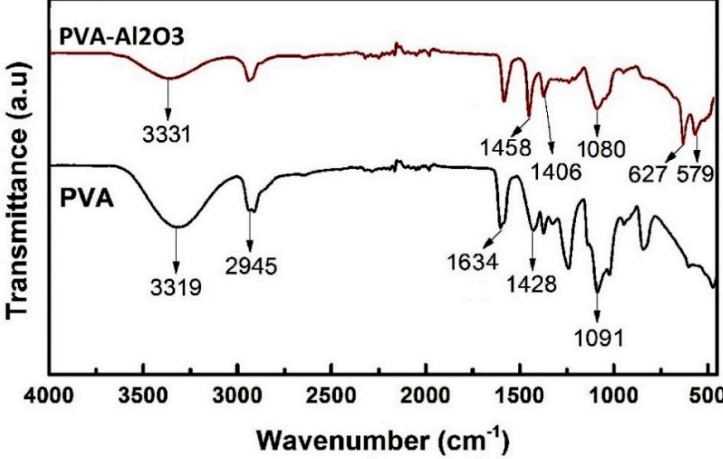

**Figure 2.** FTIR of original PVA and PVA-Al$_2$O$_3$ (10 wt%) nanocomposite.

2.1.2. Field Emission Scanning Electron Microscope (FESEM) and Energy Dispersive X-ray (EDX) Analyses

FESEM was applied to study the morphological changes in PVA with the incorporation of $Al_2O_3$ molecules. FESEM micrographs of the virgin PVA, $Al_2O_3$ nanoparticles, and the hybrid PVA-$Al_2O_3$ composite are shown in Figure 3. The surface of the unmodified PVA (Figure 3a) was obtained to be smooth and homogeneous with great extent as compared to the hybrid composite film (Figure 3c). As shown in the micrographs, the surface of polymer was highly deformed upon metal oxide interaction and the metal oxide particles appeared as bright spots that were homogeneously distributed throughout the polymer surface, which has a great similarity to the morphological surface of $Al_2O_3$ nanoparticles as shown in Figure 3b. Moreover, energy dispersive X-ray (EDX) measurements for the modified PVA-$Al_2O_3$ composite film confirmed the presence of alumina inside the PVA matrix. From the EDX results, the elemental analysis of modified composite determined the $Al_2O_3$ content in the matrix to be 10 wt% (Figure 4).

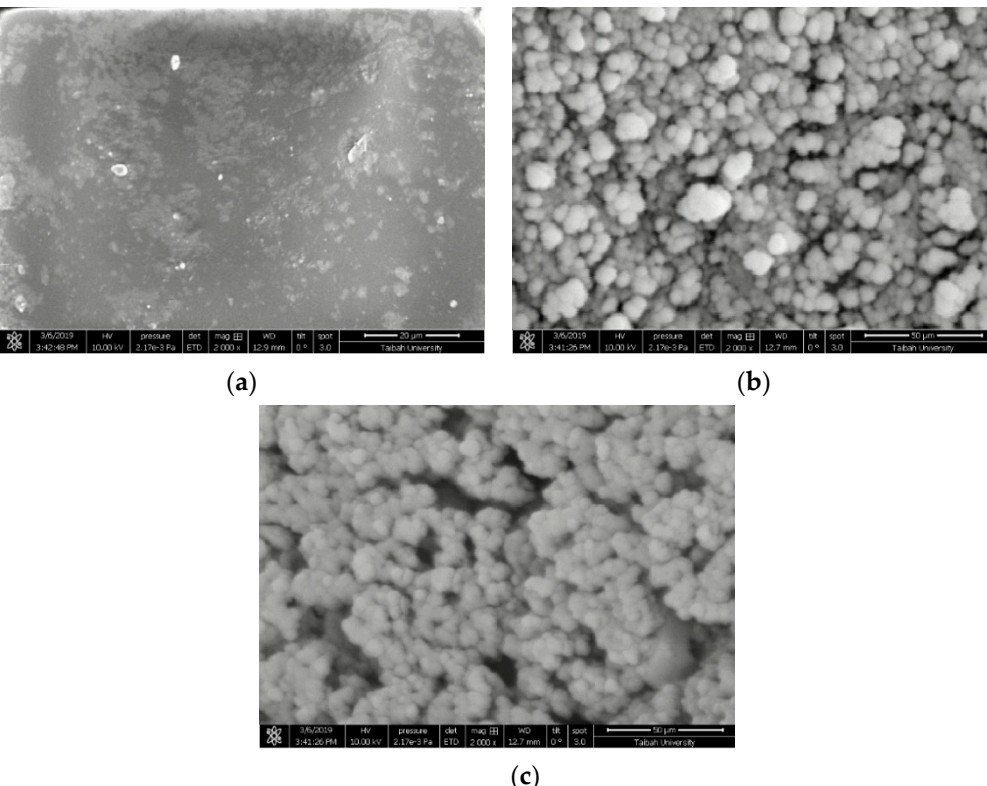

**Figure 3.** (**a**) SEM image of PVA; (**b**) SEM image of $Al_2O_3$ nanoparticles; and (**c**) SEM image of PVA/$A_2O_3$ film.

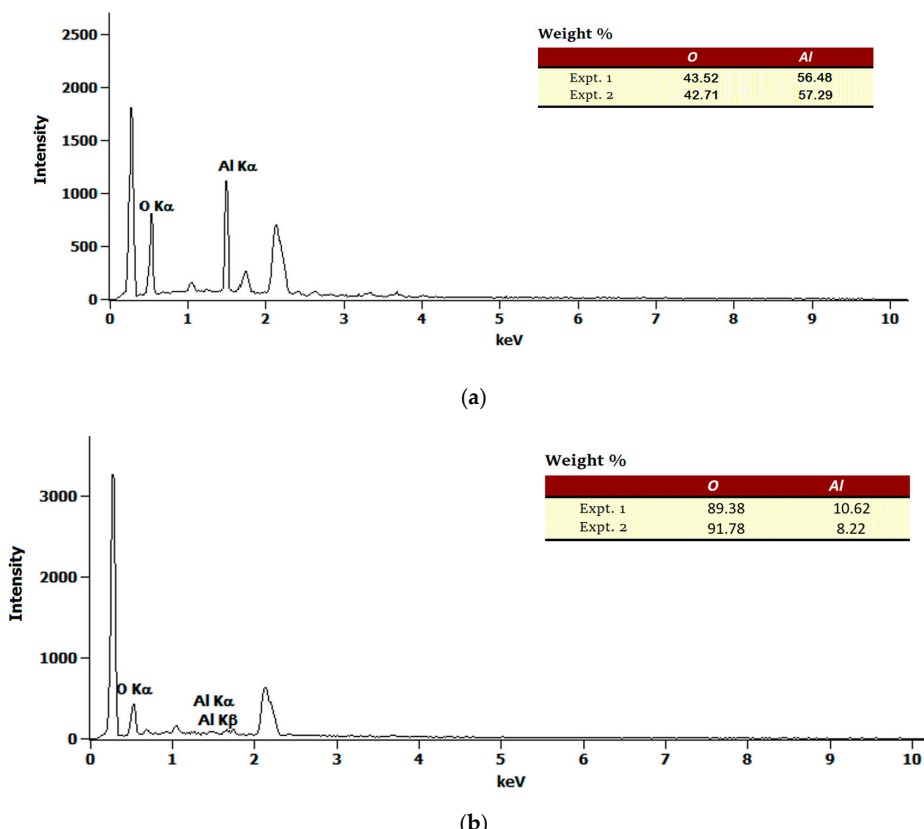

(a)

(b)

**Figure 4.** (**a**) energy dispersive X-ray (EDX) spectra of Al$_2$O$_3$ nanoparticles and (**b**) EDX spectra of PVA-Al$_2$O$_3$ (10 wt%).

### 2.1.3. Structure Analysis

X-ray diffraction (XRD) patterns of the control PVA (obtained under the same conditions but in absence of alumina powder) and the PVA-Al$_2$O$_3$ thin film composite were shown in Figure 5. The pattern of control PVA contains only a characteristic sharp peak at 2 θ = 20° as reported in literature [20]. For the PVA-Al$_2$O$_3$ composite pattern there was a clear broad hump that indicates the majority of the amorphous nature of virgin PVA. It is obviously seen that the relatively higher sharpness of the diffraction peaks a bit with the incorporation of Al$_2$O$_3$ particles, which indicates the interaction with alumina results in a clear decrease in the PVA crystallinity. Moreover, the disappearance of the characteristic Al$_2$O$_3$ peaks (ASTM 75-1862) and the lower crystallinity of the PVA-Al$_2$O$_3$ nanocomposite are attributed to the strong interaction between Al$_2$O$_3$ and PVA [16].

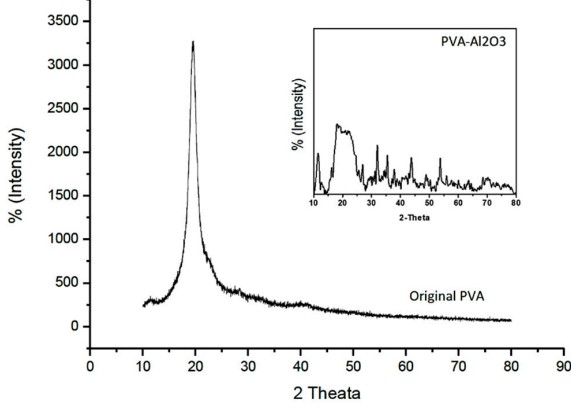

**Figure 5.** XRD pattern of virgin PVA and PVA-Al$_2$O$_3$ nanocomposite.

## 2.2. Synthesis of Thiazole Derivatives

The reaction conditions between thiosemicarbazones and $\alpha$-haloketones or $\alpha$-haloesters were optimized based on the model reaction of 2-benzylidenehydrazine-1-carbothioamide (**1a**) and ethyl 2-chloro-3-oxobutanoate (**2a**) to estimate the proper catalytic loading and time (Scheme 1 and Table 1).

**Scheme 1.** Synthesis of **3a** as a model product.

**Table 1.** Optimization of catalytic loading of PVA-Al$_2$O$_3$ nanocomposite as a basic catalyst.

| Entry | Loading Catalyst (Wt%; *) | Time (min) | Yield (%) | Entry | Loading Catalyst (Wt%; *) | Time (min) | Yield (%) |
|---|---|---|---|---|---|---|---|
| 1 | 2.5 | 60 | 40 | 7 | 10 | 30 | 40 |
| 2 | 5 | 60 | 50 | 8 | 10 | 60 | 60 |
| 3 | 10 | 60 | 60 | 9 | 10 | 90 | 72 |
| 4 | 15 | 60 | 60 | 10 | 10 | 120 | 80 |
| 5 | 20 | 60 | 59 | 11 | 10 | 150 | 87 |
| 6 | 25 | 60 | 56 | 12 | 10 | 180 | 92 |

(*) Method A, by using 10 wt% of PVA/Al$_2$O$_3$ nanocomposite film as basic catalyst.

As shown in Table 1, the reactions were proceeded in dioxane under thermal conditions with a loading weight percentage of (2.5%, 5%, 10%, 15%, 20%, and 25%) PVA-Al$_2$O$_3$ nanocomposite as a basic catalyst. The results indicated that, the optimal loading catalyst was 10 wt% (entry 3) for the reaction time 60 min. In addition, the reactions were screened to estimate the optimal time. It was noted that, the formation of thiazole product **3a** was significantly increased with time (entries 7–12). Thus, the maximum yield percent was obtained after 180 min (entry 12), which produced the product **3a** with 92% yield. Moreover, the recyclability of PVA-Al$_2$O$_3$ nanocomposite as a basic catalyst was also investigated. The catalyst was reused three times without significant loss of its catalytic efficiency (Table 2) under optimum conditions (10 wt% and 180 min).

**Table 2.** Recyclability of PVA-Al$_2$O$_3$ nanocomposites as basic catalyst.

| State of Catalyst | Fresh Catalyst | Recycled (1) | Recycled (2) | Recycled (3) |
|---|---|---|---|---|
| Yield (%) of product 3a (*) | 92 | 91 | 90 | 90 |

(*) Method A, by using 10 wt% of PVA/Al$_2$O$_3$ nanocomposite film as basic catalyst.

Subsequently, a wide array of 2-arylidenehydrazine-1-carbothioamides (**1a–g**), bearing electron-donating or electron-withdrawing group, and $\alpha$-haloesters **2a** or $\alpha$-haloketones **2b** were refluxed in dioxane for 3h, under different conditions, and provided 2-hydrazinothiazoles **3a–j** (Scheme 2 and Table 3).

**Scheme 2.** Synthesis of thiazole derivatives **3a**–**j**.

**Table 3.** Yield percent of products **3a**–**j**.

| Compd. No. | $R^1$ | $R^2$ | $R^3$ | $R^4$ | Yield (%) | | | M.P. (°C) | Ref. |
|---|---|---|---|---|---|---|---|---|---|
| | | | | | No Catalyst | $Al_2O_3$ (*) | PVA/$Al_2O_3$ (**) | | |
| 3a | H | H | H | COOEt | 78 | 90 | 92 | 195–197 | [21] |
| 3b | H | H | Cl | COOEt | 77 | 87 | 90 | 205–207 | [21] |
| 3c | H | H | NO$_2$ | COOEt | - | - | 94 | 260–262 | [21] |
| 3d | H | H | N(CH$_3$)$_2$ | COOEt | - | - | 95 | 230–232 | [21] |
| 3e | H | H | OCH$_3$ | COOEt | - | - | 95 | 180–182 | [21] |
| 3f | Cl | H | Cl | COOEt | - | - | 92 | 222–224 | [21] |
| 3g | H | OCH$_3$ | OH | COOEt | - | - | 97 | 226–228 | [21] |
| 3h | H | H | H | COCH$_3$ | - | - | 93 | 222–224 | [22] |
| 3i | H | H | OCH$_3$ | COCH$_3$ | - | - | 91 | 214–216 | [22] |
| 3j | Cl | H | Cl | COCH$_3$ | - | - | 96 | 240–242 | [22] |

(*) Method B in experimental part. (**) Method A in experiment part.

As shown in Table 3, the comparative study for the yield percentage of the products **3a** and **3b** under different reaction conditions, namely blank experiments (without catalyst), control experiments ($Al_2O_3$ nanoparticles), and PVA/$Al_2O_3$ nanocomposite film was investigated. The yield was enhanced in the presence of the later basic catalysts. Although the catalytic performance of both basic catalysts were quite similar, PVA/$Al_2O_3$ nanocomposite film was more superior due to the ease of its recyclability and also a relatively smaller amount of $Al_2O_3$ was required in the catalyst preparation.

A plausible mechanism for synthesis of 2-hydrazinothiazoles **3a**–**j** using PVA/$Al_2O_3$ nanoparticles was depicted in Scheme 3.

**Scheme 3.** Plausible mechanism for synthesis of 2-hydrazinothiazoles **3a**–**j**.

$Al_2O_3$ nanoparticles act as a Bronsted base [11] via deprotonation of the thiol group of 2-arylidenehydrazine-1-carbothioamide (**1a**–**g**). The producing thiolate anion attacks $\alpha$−halocarbonyl

compounds (2a,b) with displacement of chlorine atom to give non-isolable intermediate. Cyclocondensation of the latter intermediate afforded the authenticated thiazole derivatives **3a–j**.

## 3. Materials and Methods

PVA (molecular weight = 22,000 g/mol, density = 1.19 g/cm$^3$), Al$_2$O$_3$ nanoparticles (Merck 642991; 30–60 nm particle size (TEM); 20 wt% in water), and ethylenediamine (ED) were purchased by Sigma Aldrich. Thiosemicarbazones **1a–g** were prepared as reported in literature [23–27]. Melting points of authenticated samples **3a–j** were measured on an electrothermal Gallenkamp capillary apparatus (Leicester, UK). The FTIR spectra of nanocomposite were recorded on a Pye-Unicam SP300 Instrument (Cambridge, UK) in potassium bromide discs. FESEM analyses were measured on a high resolution scanning electron microscope (model HRSEM, JSM 6510A, Jeol, Tokyo, Japan). XRD measurement was carried out on Philips Diffractometer (Model: X'Pert-Pro MPD; Philips, Eindhoven, The Netherland).

### 3.1. Preparation of PVA-Al$_2$O$_3$ Composite Films

The PVA and Al$_2$O$_3$ hybrid composite was prepared using the simple solution casting method. Initially, 0.5 g of PVA was dissolved in deionized water (50 mL) under stirring for 6 h at 80 °C then, 0.1 g of Al$_2$O$_3$ nanoparticles was added to the latter PVA solution and the mixture was kept under magnetic stirring for 1 h till a dispersion viscous solution was formed. The produced viscous solution was casted in a pre-cleaned Teflon Petri-dish and this solution was annealed in an oven adjusted at 150 °C for 1 h. The final PVA-Al$_2$O$_3$ films for the study was 125 mm × 12 mm × 1.5 mm in dimensions. Furthermore, a pure PVA film, without Al$_2$O$_3$, was prepared in a similar way for comparative studies (Figure 6).

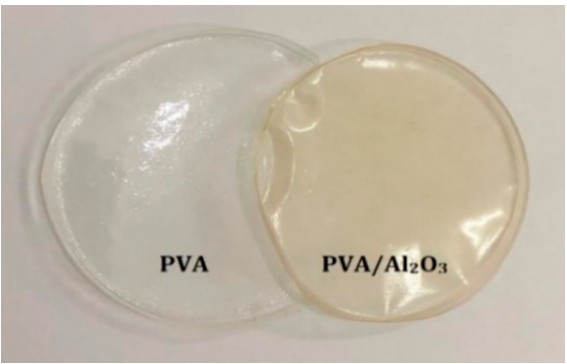

**Figure 6.** Images of virgin PVA and PVA-Al$_2$O$_3$ nanocomposite.

### 3.2. General Procedures for the Synthesis of 2- Hydrazonothiazoles **3a-j**

Method A: A mixture of thiosemicarbazones **1a–g** (1 mmol) in dry dioxane (20 mL), containing PVA/Al$_2$O$_3$ film (0.1 g, 10% wt relative to thiosemicarbazones), and ethyl 3-chloro-2-oxobutanoate (**2a**) or 3-chloro-2,4-pentanedione (**2b**) (1 mmol of each) was refluxed until all the starting material was consumed (3 h. as monitored by TLC). After completion of the reaction, the hot solution was filtered to remove the film and the filtrate was poured into 50 mL ice/50 mL, 6 M HCl mixture. The precipitate was filtered, washed with water, and crystallized from appropriate solvent to give authenticated thiazole derivatives **3a–j** [21,22]. The film was washed with hot ethanol and dried in an oven adjusted at 150 °C for 1 h to be reused following the same procedure in method A.

Method B: the same procedure in method A was applied using Al$_2$O$_3$ nanoparticles instead of PVA-Al$_2$O$_3$ film. After the completion of the reaction, the solution was poured into 50 mL ice/50 mL, 6 M HCl mixture to get rid of Al$_2$O$_3$ nanoparticles.

## 4. Conclusions

In this work, a new hybrid PVA/Al$_2$O$_3$ nanocomposite was prepared via simple solution casting method and the obtained hybrid nanocomposite film exhibited an obvious catalytic performance for synthesis of 2-hydrazonothiazoles with an excellent yield. The optimal loading catalyst was 10 wt% and the catalyst was reused effectively three times without significant loss of its catalytic efficiency.

**Author Contributions:** Both S.M.R. and K.D.K. suggested the plan of the article. K.D.K. interpreted the results of preparation of catalyst, and S.M.R. interpreted the chemistry part. A.H.B. participated in the reviewing and publication processes of the article. All authors have read and agreed to the published version of the manuscript.

**Funding:** This research received no external funding.

**Conflicts of Interest:** The authors declare no conflict of interest.

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
