# Peer review of "Structural Properties and Catalytic Activity of Binary Poly (vinyl alcohol)/Al2O3 Nanocomposite Film for Synthesis of Thiazoles"

_catalysts, doi:10.3390/catal10010100_

Round 1

Reviewer 1 Report

M. Riyadh et al. describes a catalytic performance of poly (vinyl alcohol)/Al2O3 nanocomposite film. This type of hybrid organic-inorganic materials is of a high interest from practical and academic points of view. However, before the publication in “Catalysts” the following issues should be addressed.

Introduction should highlight the novelty of the paper. If the novelty relates to the synthesis procedure for PVA/Al2O3 nanocomposite preparing, it should be discussed in Results and Discussion section. Introduction should be supplemented by the literature data on other PVA/Al2O3 composites. SEM images on the Fig. 3 are blurry and scales are poorly visible. The proposed mechanism for synthesis of 2-hydrazinothiazoles over PVA/Al2O3 should be supported by the experimental data. The column 3 in the Table 1 should be revised, because its presents identically values, the same for the column 6. The identification of the produced thiazole derivatives, first at all by NMR, should be presented in the Experimental part. The acid sites of Al2O3 in the nanocomposite should be evaluated by IR using acetonitrile or pyridine as probe molecules. Generally, the paper is rather compendious and should be supplemented by an additional discussion of the obtained results.

Author Response

Ms. Cora Huang, M.Sc.

Assistant Editor - Catalysts

Thank you for your E-mail concerning manuscript ID (catalysts-681847-layout) entitled "Structural properties and catalytic activity of binary poly (vinyl alcohol)/Al2O3 nanocomposite film for synthesis of thiazoles"

Please find attached our revised manuscript having complied with linguistic as well as technical remarks of referees.

With respect to remarks of referee's comment-1

Introduction should highlight the novelty of the paper. If the novelty relates to the synthesis of procedure for PVA/Al2O3 nanocomposite preparing, it should be discussed in Results and Discussion section.

Some information about the metal oxide/polymer composites and the novelty of the prepared PVA/Al2O3 is added to the introduction section in the article. In addition, characterization of the nanocomposite by studying FTIR, SEM, and XRD was shown in the Results and Discussion section.

Introduction should be supplemented by the literature data on other PVA/Al2O3

Two references are added to the introduction section referring to the previously published data for other PVA/Al2O3 composites for another applications.

SEM images on the Fig. 3 are blurry and scales are poorly visible.

The poor resolution SEM graphs are replaced by more resolution ones and the SEM of the nanosized Al2O3 are added.

The proposed mechanism for synthesis of 2-hydrazinothiazoles over PVA/Al2O3should be supported by the experimental data.

The proposed mechanism is suggested on the basis of the similar mechanism that was previously published in Ref. 11.

The column 3 in the Table 1 should be revised, because its presents identically values, the same for the column 6.

Column 3 (in Table 1) means the studying of the effect of catalyst loading on the Yield% under constant time. Similarly, Column 6 refers to the relation between Time and Yield% under constant catalyst loading.

The identification of the produced thiazole derivatives, first at all by NMR, should be presented in the Experimental part.

The investigated catalytic reactions are previously confirmed and published (References are added to refer their preparations).

The acid sites of Al23in the nanocomposite should be evaluated by IR using acetonitrile or pyridine as probe molecules.

Al2O3 is known as an amphoteric substance but in presence of substrates with acidic protons it acts as Bronsted base that deprotonates the acidic substrates that is why it is used effectively in these base catalyzed reactions.

I sincerely hope that the amended version is satisfactory for publication.

Thanking you for your cooperation and best regards.

Sincerely, yours

Dr. Khaled D. Khalil

Reviewer 2 Report

In this work, authors tried to use PVA/Al2O3 for synthesis of thiazoles.

I have comments regarding the content of their manuscript as follows.

This manuscript is too short and few data have been shown. They should provide more physical and chemical data (catalyst loading, morphology structure, diffraction analysis etc.) of the spent catalysts after 1 h or 3 h of reaction and catalyst stability for longer reaction time. They should incorporate these data in the results and discussion section. Lines 89,95,96 and Table 1

        If the reaction increase with increasing reaction time, could the authors obtain 100% yield after more than 200 min reaction time?

        Do the authors have yield data after 5 h or 6 h? (see comment No. 1).

Line 100, Table 2, Authors, please provide recyclability of Al2O3. I suppose they have the same degree of recyclability with that of PVA/Al2O3. Lines 108, 113-115 and Table 3

        It is difficult to see the effect of PVA on Al2O3 as they have almost       similar yield. There is no significant increase in the yield after PVA was incorporated into Al2O3.

Author Response

Ms. Cora Huang, M.Sc.

Assistant Editor - Catalysts

Thank you for your E-mail concerning manuscript ID (catalysts-681847-layout) entitled "Structural properties and catalytic activity of binary poly (vinyl alcohol)/Al2O3 nanocomposite film for synthesis of thiazoles"

Please find attached our revised manuscript having complied with linguistic as well as technical remarks of referees.

With respect to remarks of referee's comment-2

This manuscript is too short and few data have been shown.

Some information about the metal oxide/polymer and the novelty of PVA/Al2O3 is added to the introduction section in the article.

They should provide more physical and chemical data (catalyst loading, morphology structure, diffraction analysis etc.) of the spent catalysts after 1 h or 3 h of reaction and catalyst stability for longer reaction time. They should incorporate these data in the results and discussion section. Lines 89,95,96 and Table 1.

Actually, the requested data were measured for authenticated sample of the used catalyst and the results were in agreement with our conclusion that there is no marked change in the surface and catalytic activity of the proposed catalyst even after its recovering (See Table 2).

If the reaction increase with increasing reaction time, could the authors obtain 100% yield after more than 200 min reaction time? Do the authors have yield data after 5 h or 6 h? (See comment No. 1).

From the results of the catalytic reactions, the Yield% of the products was regularly increased with increasing the allowed time for the reaction till 3 h, where it was constancy (in the Yield%) that is why we are satisfied with maximum time of 3 h.

Line 100, Table 2, Authors, please provide recyclability of Al2O3. I suppose they have the same degree of recyclability with that of PVA/Al2O3. Lines 108, 113-115 and Table 3.

Al2O3 nanoparticles was purchased from Sigma-Aldrich [Merck 642991; 30-60 nm particle size (TEM); 20 wt. % in water], its recovering from the reaction media is not easy and leads to a contamination of the product that is why we prefer to use it as nanoparticles incorporated within the PVA matrix film. So, its removal and recovering will be easier.

It is difficult to see the effect of PVA on Al2O3 as they have almost similar yield. There is no significant increase in the yield after PVA was incorporated into Al2O3.

In this work, the main role of PVA is its utility as a polymer matrix in which Al2O3 nanoparticles are distributed. From the economic point of view, the composite is preferred where relatively small amount of Al2O3 is required in the catalyst preparation, effective catalytic potency, and ease of recovering (without loss in its activity) are all promising.

In Result and discussion section, we stated that "Although the catalytic performance of both basic catalysts are quite similar, PVA/Al2O3 nanocomposite film more superior due to the ease of its recyclability".

I sincerely hope that the amended version is satisfactory for publication.

Thanking you for your cooperation and best regards.

Sincerely, yours

Dr. Khaled D. Khalil

Round 2

Reviewer 1 Report

Authors have revised the Manuscript according to referee comments, 

so it can be published in the present form.

Reviewer 2 Report

I have read the revised manuscript. Authors have addressed my concerns.

I think this revised manuscript is acceptable to be published in the Catalysts.